# Recent Insights into Human Endometrial Peptidases in Blastocyst Implantation via Shedding of Microvesicles

**DOI:** 10.3390/ijms222413479

**Published:** 2021-12-15

**Authors:** Masato Yoshihara, Shigehiko Mizutani, Yukio Kato, Kunio Matsumoto, Eita Mizutani, Hidesuke Mizutani, Hiroki Fujimoto, Satoko Osuka, Hiroaki Kajiyama

**Affiliations:** 1Department of Obstetrics and Gynecology, Nagoya University Graduate School of Medicine, 65 Tsuruma-cho, Showa-ku, Nagoya 466-8550, Japan; mizutani.hide@med.nagoya-u.ac.jp (H.M.); fujimoto.hiroki@med.nagoya-u.ac.jp (H.F.); satokoosuka@med.nagoya-u.ac.jp (S.O.); kajiyama@med.nagoya-u.ac.jp (H.K.); 2Daiyabilding Lady’s Clinic, 1-1-17 Meieki, Nishi-ku, Nagoya 451-0045, Japan; daiya-lc@amber.plala.or.jp; 3Department of Molecular Pharmacotherapeutics, Faculty of Pharmacy, Kanazawa University, Kanazawa 920-1192, Japan; ykato@p.kanazawa-u.ac.jp; 4Division of Tumor Dynamics and Regulation, Cancer Research Institute, Kanazawa University, Kanazawa 920-1192, Japan; kmatsu@staff.kanazawa-u.ac.jp; 5Discipline of Obstetrics and Gynaecology, Adelaide Medical School, Robinson Research Institute, University of Adelaide, Adelaide, SA 5005, Australia

**Keywords:** extracellular vesicles, lysosome, dipeptidyl peptidase IV, aminopeptidase N, blastocyst implantation

## Abstract

Blastocyst implantation involves multiple interactions with numerous molecules expressed in endometrial epithelial cells (EECs) during the implantation window; however, there is limited information regarding the molecular mechanism underlying the crosstalk. In blastocysts, fibronectin plays a major role in the adhesion of various types of cells by binding to extracellular matrix proteins via the Arg-Gly-Asp (RGD) motif. In EECs, RGD-recognizing integrins are important bridging receptors for fibronectin, whereas the non-RGD binding of fibronectin includes interactions with dipeptidyl peptidase IV (DPPIV)/cluster of differentiation (CD) 26. Fibronectin may also bind to aminopeptidase N (APN)/CD13, and in the endometrium, these peptidases are present in plasma membranes and lysosomal membranes. Blastocyst implantation is accompanied by lysosome exocytosis, which transports various peptidases and nutrients into the endometrial cavity to facilitate blastocyst implantation. Both DPPIV and APN are released into the uterine cavity via shedding of microvesicles (MVs) from EECs. Recently, extracellular vesicles derived from endometrial cells have been proposed to act on trophectoderm cells to promote implantation. MVs are also secreted from embryonal stem cells and may play an active role in implantation. Thus, crosstalk between the blastocyst and endometrium via extracellular vesicles is a new insight into the fundamental molecular basis of blastocyst implantation.

## 1. Introduction

The human endometrium consists of a single layer of columnar epithelium, which comprises endometrial epithelial cells (EECs), and a stroma, which comprises endometrial stromal cells (ESCs). During the normal menstrual cycle, the endometrium undergoes cyclic changes, and ovarian steroid hormones are considered to control the differentiation and function of the endometrium. The endometrium is composed of basal and functional layers, the latter comprising the superficial two-thirds of the endometrium.

Shortly after ovulation, progesterone increases the permeability of the plasma membranes and lysosomal membranes of EECs; the enhanced permeability leads to the secretion of various peptidases and other nutrients into the endometrial cavity (Figure 1), possibly facilitating blastocyst implantation within the limited period of the implantation window (days 19 to 21 of menstrual cycles) [1]. Thus, EECs are profoundly dynamic as the site of blastocyst implantation. In recent years, emerging evidence has indicated that extracellular vesicles (EVs), such as exosomes and microvesicles (MVs), produced by cells are involved in intercellular communication. Increasing evidence has also shown that EVs of maternal and/or embryonal origin participate in the blastocyst–endometrial interactions that are critical to implantation (Figure 1) [2,3].

Although the mechanisms underlying the molecular basis of crosstalk during the implantation window are largely unknown, fibronectin is known to play a central role in the adhesion of various types of cells. Fibronectin interacts with the cell surface via the Arg-Gly-Asp (RGD) motif, RGD-recognizing proteins, and primarily integrin, which has been suggested to be an important bridging ligand of fibronectin in blastocysts during implantation. On the other hand, fibronectin is also involved in implantation via non-RGD binding, and the fundamental role of blastocyst fibronectin in the implantation process has been further supported by the identification of large-sized molecules in EECs that bind to fibronectin via a non-RGD motif [4].

In this review, the roles of RGD- and non-RGD-containing proteins that possibly interact with fibronectin in blastocyst implantation are discussed. One example of such proteins is dipeptidyl peptidase IV (DPPIV)/cluster of differentiation (CD) CD26, which is highly expressed in glandular epithelium during the implantation window [5], and is involved in the interaction with blastocyst fibronectin during implantation [6]. The adhesion mechanism of fibronectin and DPPⅣ/CD26 is different from that of the RGD motif. Aminopeptidase N (APN)/CD13 also binds to blastocyst fibronectin via the Asn-Gly-Arg (NGR) motif, and is expressed in ESCs of the endometrium. Both DPPIV and APN are also present in EVs and lysosomes; therefore, we proposed the hypothesis that lysosomes are secreted as EVs, and blastocyst fibronectin may also bind to these peptidases localized in EVs/lysosomes on the outer side of the endometrium. The blastocyst attached to the EVs/lysosomes may subsequently return to the endometrium, leading to completion of blastocyst implantation, wherein glycocalyx plays a central role as the destination of the blastocyst attached to the EVs/lysosomes [7,8]. Lastly, the possible involvement of blastocysts in implantation is discussed; EVs secreted from embryonic stem (ES) cells of a preimplantation embryo have been shown to play active roles in implantation [9].

## 2. Endometrial Change during Decidualization

Epithelial–mesenchymal transition (EMT) is a critical process in various developmental stages. The essential step in EMT is the loss of epithelial cell markers; a notable example is the decreased expression of E-cadherin [10]. E-cadherin is responsible for maintaining the lateral contacts of epithelial cells via adherent junctions, cell adhesion, and relative immobility in tissues [11]. Matsuzaki et al. [12] showed that a significantly higher expression of E-cadherin was detected in infertile patients with endometriosis, while its expression was extremely low or not detected in the EECs of the mid-secretory endometria of healthy fertile women. Progesterone may play a key role in maintaining the epithelial phenotype by actively inhibiting EMT, as demonstrated using various cell models [13,14,15], whereas the temporal downregulation or loss of E-cadherin expression in EECs during the implantation window may be necessary for blastocyst implantation.

### 2.1. Changes in ESCs during Decidualization

Human endometrial decidualization begins approximately 6 d after ovulation at the onset of the implantation window. The stromal compartment of the human endometrium becomes edematous, with an increase in ESC cytoplasm; this change is initiated in ESCs near the terminal spiral arteries, and subsequently expands throughout the stroma over the secretory phase of the menstrual cycle [1]. Although the reason for the initiation of decidualization around the terminal spiral arteries is unknown, various molecules exudated from the blood vessels may contribute to the initiation of decidualization at the end of the arteries. These changes in ESCs are essentially parallel to the changes in EECs during decidualization; various changes observed in ESCs during decidualization, such as those in intracellular organelles and the accumulation of glucose and various nutrients, are similarly observed in EECs.

The substantial differentiation process of the uterine endometrium during decidualization includes both morphological and biochemical changes aimed at blastocyst implantation. Decidualization is a biological transformation process that closely resembles a mesenchymal–epithelial transition that occurs independent of the presence of an implanting blastocyst; ESCs are differentiated from elongated fibroblast-like ESCs, into cells with a more rounded and highly specialized secretory epithelioid cell type, termed decidual cells [16]. Generally, the mesenchymal phenotype exhibits a distinct elongated cell shape with anterior–posterior polarity and the ability to move in the extracellular matrix, rather than attaching to it via a leading pseudopodium extruded from a continuously formed plasma membrane [16].

Decidualizing ESCs also undergo biochemical alterations, including an expansion of the rough endoplasmic reticulum and Golgi complex, accumulation of glycogen and lipid droplets in the cytoplasm, enhanced expression of certain extracellular matrix proteins, such as laminin and type IV collagen, and an increase in the production of secretory proteins, including prolactin and insulin-like growth factor binding protein 1 (both of which are well-known markers of decidualization) [17,18]. Thus, decidualization in the endometrium is profound and gradually affects all uterine compartments.

Another important aspect of decidual cells is that these cells lack aminopeptidase A (APA), the AP responsible for the degradation of angiotensin II (A-II). Vasoconstrictive neuropeptides, such as A-II, arginine vasopressin, and oxytocin, play critical roles in embryo growth [1]. The absence of APA in the decidual cell area may constitute a critical preparation for implantation; A-II shows a growth-factor-like effect at 10^−11^ M on post-implantation embryos in rats, and expression of A-II type 1 and 2 receptors (AT1R and AT2R, respectively) in the preimplantation embryos suggests that the embryos are sensitive to A-II, which is present in the early gestational environment [19,20]. The lack of APA in the decidual cell area supports A-II as a growth factor in postimplantation human embryos. Once an embryo reaches the uterus, it first encounters decidual cells that lack APA. A-II, presumably supplied from the maternal endometrium, reaches the embryo without being exposed to degradation by APA. The lack of APA in the decidual cell area in the endometrium can strengthen A-II as a growth-factor-like effector after implantation. This may be compatible with A-II-enhanced glucose uptake in ES cells [21].

### 2.2. Changes in EECs during Decidualization

#### 2.2.1. Apocrine and Holocrine Secretions from EECs during the Implantation Window

EECs are essentially epithelial phenotypes, whereas downregulation of the epithelial marker E-cadherin seems to be essential for implantation, as mentioned in Section 2. After ovulation, the human endometrium actively secretes glycoproteins, including aminopeptidase (APs) and nutrients, such as peptides and amino acids, into the endometrial cavity. The secretion is mediated by exocytosis of lysosomes, which is stimulated by progesterone (Figure 1) to facilitate blastocyst implantation. Progesterone is also involved in the influx of blood-derived glucose and other nutrients into EECs. The exocytosis observed in this process is a drastic phenomenon, such as apocrine and/or holocrine secretion, and reaches its maximal level within the implantation window. Apocrine secretion from EECs is believed to serve as communication with the exterior fertilized ovum by the transfer of cellular products [1].

After ovulation, intracellular Ca^2+^ and cyclic monophosphate (cAMP) levels are elevated in EECs and ESCs, by surges in estrogen and luteinizing hormone levels. In addition, elevated Ca^2+^ is known to promote lysosomal membranes to fuse with other cellular membranes, resulting in the exocytosis of lysosomal contents (Figure 1). Lysosomal exocytosis plays a major role in several physiological processes, such as cellular immune response, bone resorption, and plasma membrane repair, and Ca^2+^-dependent lysosomal exocytosis is observed in a variety of cell types [22,23]. The endolysosomal system constitutes a set of intracellular membranous compartments: early endosomes, late endosomes, recycling endosomes, and lysosomes (Figure 1), which can actively interconvert to each other. Lysosomes were once considered the endpoint sequence of endocytosis and degradation of macromolecules; however, they are currently recognized as dynamic organelles with significant capabilities to fuse themselves with a variety of targets, followed by subsequent postfusion reformation [24,25,26]. Lysosomes also serve as major sites for the activation of proteolytic activities of many peptidases, even if their activities can be detected in endomembrane systems other than lysosomes [26]. Various APs are postulated to activate enzymes in the endolysosomal system [1].

#### 2.2.2. Exocytosis of Lysosomes, Exosomes, and EVs in EECs

Eukaryotic cells secrete proteins produced via their biosynthetic pathways by constitutive exocytosis of secretory vesicles, and/or by the release of secretory or storage granules upon appropriate stimulation [27]. Thus, at least two different classes of EVs, namely, exosomes and MVs, were identified. Exosomes range in size from 30 to 100 nm and are derived from the rerouting of multivesicular bodies (MVBs), which are a subset of endosomes that contain membrane-bound intraluminal vesicles intended for degradation, in the lysosome to the cell surface, where they fuse with the plasma membrane and are released (Figure 1) [28]. Thus, exosomes are secreted in various cell types as a consequence of the fusion of multivesicular late endosomes/lysosomes with the plasma membrane [27]. However, the distinction between late endosomes and lysosomes is not precise, and it is speculated that MVBs belong to the category of late endosome/lysosome [27]. MVs range in size from 0.2 to 2 µm and are also referred to as ectosomes, microparticles, and oncosomes, in the case of cancer cells. MVs tend to be considerably larger than exosomes, and are formed and shed directly from the plasma membrane (Figure 1) [29]. While elevated Ca^2+^ levels cause lysosomal exocytosis, MVs are shed from the plasma membrane, depending on an increase in the population of cytosolic Ca^2+^-like lysosomes (Figure 1) [30]. The exocytosis of lysosomes contributes to the repair of the plasma membrane [22]; therefore, lysosomes probably constitute a part of MVs.

The pancreatic exocrine acinar cells transport lysosomal enzymes from the Golgi apparatus to the acinar lumen, suggesting the direct exocytosis of lysosomes [31]. Similarly, we have postulated the progesterone-induced apocrine secretion of lysosomes, i.e., the direct exocytosis of lysosomes in EECs [1]. The size of lysosomes varies from 0.1 to 1.2 μm, with the largest being more than 10 times that of the smallest. The size of lysosomes in EECs increases dramatically after ovulation [1].

## 3. Implantation

### 3.1. Fibronectin in Blastocysts

After the egg is successfully fertilized in the fallopian tube, the egg travels through the fallopian tube toward the uterus, during which the fertilized egg divides and develops into a multicellular structure called a blastocyst. Following migration and shedding of the zona pellucida, the trophectoderm of the implanting blastocyst must be prepared for implantation prior to attachment to the maternal endometrium. Blastocysts are composed of an inner cell mass (comprising ES cells) and the surrounding epithelial trophectoderm.

Integrins are expressed in the endometrium and are suggested to be involved in implantation [32]. The ligands for integrins include fibronectin, which is expressed in the early embryo [33]. Our previous study showed that fibronectin mRNA is detected at the blastocyst stage, but not at the morula stage in human embryos, and its existence on the cell surface in human hatched blastocysts was verified via immunefluorostaining [6]. The interaction of various integrins with large proteins in the maternal endometrial epithelium may mediate the adhesion of blastocysts [34]. This may be compatible with the previous finding that the trophectoderm in humans may initially attach to the glycocalyx in the maternal endometrial epithelium, in which adhesion molecules, such as integrins, are highly expressed [7]. Thus, fibronectin in blastocysts probably plays an essential role in implantation by binding to large molecules in EECs.

### 3.2. Dynamic Aspects of EECs during the Implantation Window and Their Relation to APs

The surface areas of EECs show dynamic changes, mainly with rest to apocrine secretion, in addition to lysosomal exocytosis, which are considered to serve as communications with the exterior fertilized ovum via the transfer of cellular products [1]. Scanning electron microscopy confirmed that during the implantation window, the epithelia of both surface and glandular EECs are composed of four cell types: microvilli-rich cells, pinopode cells, vesiculated cells, and ciliated cells. Dynamic vesicular transport, which includes endocytosis, transcytosis, and exocytosis, was evident in these cells, implying that they actively communicate with the external environment and neighboring cells.

The surface and glandular epithelia in pinopode cells are morphologically different; on the surface epithelium, pinopodes occupy the entire luminal cell surface, with poor cytoplasm in organelles, whereas in the glandular epithelium, pinopodes tend to be thinner and irregularly shaped, with their cytoplasm frequently filled with secretory material. In addition, the glandular epithelium in pinopode cells showed apocrine secretion, which is scarce in the surface epithelium. Such differences in morphological features may be associated with different functions between the surface and glandular pinopodes, while the surface pinopodes appear to be important in blastocyst implantation, and the glandular pinopodes participate in secretion.

Pinopode cells are characterized by large cytoplasmic apical protrusions, whereas the cytoplasm of pinopodes is composed of coated vesicles, secondary lysosomes, lipid droplets, and glycogen aggregates. This morphology might be consistent with the secretion features of EECs that exhibit merocrine, apocrine, and holocrine secretions during the implantation window [35]. The elevated intracellular Ca^2+^ and cAMP levels stimulate exocytosis of the lysosomal contents, which is accompanied by an increase in progesterone, leading to an increase in the permeability of both lysosomal and cell membranes; moreover, exocytosis of lysosomes is exaggerated, resulting in apocrine and holocrine secretions in EECs [1]. In the liver, lysosomal enzymes are discharged into the bile canaliculus after glucagon stimulation. This process is accompanied by an increase in the pericanalicular distribution of secondary lysosomes, as observed in pinopode cells [36].

### 3.3. Interaction of Blastocyst Fibronectin with the Molecules in EVs/Lysosomes Derived from EECs

Pinopode cells in EECs are considered to constitute the site of blastocyst adhesion [37]. The cell surface in the human body is generally covered with a dense layer of glycosaminoglycans and proteoglycans, termed as the “glycocalyx” [8]. The initial attachment of the trophectoderm may occur on the glycocalyx on pinopode cells in EECs [7].

In women, EVs are present in the uterine fluid and the corresponding mucus [38], which increases the likelihood of retention and/or sequestration of EVs secreted from the luminal epithelium within the endometrial glycocalyx [39]. Importantly, proteomic analysis of EVs secreted in primary cultured EECs revealed that EVs containing proteins (EVs’ protein cargo) are regulated by estradiol and progesterone [40]. EVs were identified in the uterine fluid of women across the different phases of the menstrual cycle, suggesting that EVs are under the control of both steroidal hormones [38]. EVs’ protein cargo includes basement membrane peptidases, such as APN/CD13 and DPPⅢ, wherein the level of APN/CD13 exceeds that of DPPⅢ [40].

Pretreatment of rat blastocysts and Ishikawa cells (a well-differentiated human endometrial adenocarcinoma cell line) with RGD-blocking peptide significantly reduced blastocyst attachment to Ishikawa cells, suggesting an essential role of the interaction between RGD-containing proteins and fibronectin for implantation [4]. However, in fibronectin knockout mice, implantation appeared to be normal [41], indicating that other adhesion molecules may compensate for the role of fibronectin in blastocysts.

### 3.4. DPPIV/CD26 in Implantation

DPPIV/CD26, a 110-kDa type Ⅱ transmembrane glycoprotein that acts as a serine exopeptidase and is expressed in various epithelial tissues, is highly expressed on EECs in the implantation window [5]. Moreover, DPPIV/CD 26 is known to be a marker of the implantation-phase endometrium and accounts for a variety of regulatory processes, including glucose and chemokine homeostasis [42,43]. Another function of DPPIV/CD 26 is its ability to bind to the extracellular matrix; specifically, lung endothelial DPPⅣ/CD26 is known to be a vascular address for cancer cells decorated with cell-surface fibronectin. Fibronectin contains multiple CD26-binding sites, and CD26/DPPⅣ-binding sites in fibronectin have the following consensus motif: T(I/L)TGLX(P/R)G(T/V)X [44]. Human blastocysts efficiently adhere to DPPⅣ/CD26-overexpressing monolayer cell cultures; furthermore, DPPⅣ/CD26-mediated adhesion increased the trophectoderm spreading, suggesting that human blastocyst implantation involves an adhesion mechanism mediated by endometrial DPPIV/CD26 and embryonal fibronectin [6].

### 3.5. APN/CD13 in Implantation

APN/CD13 is a type II 150-kDa membrane metalloprotease with an extracellular-oriented catalytic domain. APN cleaves the N-terminal neutral residue of physiological peptides, and ubiquitously functions in various peptide metabolism pathways [45]. While DPPIV was detected in EECs, APN/CD13 was detected in ESCs and decidual cells [5].

Recent studies have shown that the formation of isoaspartyl residues (*iso*Asp) in integrin/fibronectin ligands by asparagine deamidation or aspartate isomerization could represent a mechanism for the regulation of integrin/fibronectin recognition. This spontaneous post-translational modification results in the NGR motif, which mimics RGD, a common integrin/fibronectin binding motif [46]. APN has been shown to mediate the intercellular adhesion process by binding to the signature NGR motif in extracellular matrix proteins and on the surface of other cells [47]. APN/CD13 also promotes β1 integrin recycling and subsequent cellular migration [48]. Given that β 1-integrin is highly expressed in the decidua [32], APN/CD13 may be involved in implantation and/or invasion of blastocysts.

Some alternative mechanisms of blastocyst attachment on the endometrial cell surface were shown as follows [49]: mucin1, a highly glycosylated polymeric protein contributes to cellular adhesive properties, which may function to facilitate blastocyst binding to the endometrial cell surface, but its action is through the L-selectin/sialyl-Lewis x adhesion system, and is not directly involved in binding between blastocyst and endometrial cells [50]. Integrins are transmembrane glycoproteins, which mediate cell-to-cell and cell-to-extracellular matrix (ECM) adhesion. The localization of ąvß3 integrin on the pinopods, on the apical surface of the luminal epithelium at the time of uterine receptivity, may suggest a role of this integrin subtype in initial implantation [51]. The interaction between fibronectin and integrin is mediated, as follows: cells bind to fibronectin through transmembrane receptor proteins of the integrin family, which mechanically couple the actin cytoskeleton to the ECM via an elaborate adhesion complex. Therefore, fibronectin is essential for understanding the role of integrin. Osteopontin is a glycoprotein produced by endometrial epithelia and secreted into the uterine lumen at the time of implantation, where it binds to the ąvß3 integrin present on the surface of uterine luminal epithelia [52]. Osteopontin is an integrin-binding secreted protein that contains an Arg-Gly-Asp (RGD) amino acid sequence, and binds to various cell types via RGD-mediated interaction with the ąvß3 integrin [53]. Osteopontin is present in the glycocalyx, apical cytoplasmic vesicles, and multicompartment granules in human gastric mucous cells. In human gastric mucosa, the localization of osteopontin in subcompartment granules may be important for the secretion of osteopontin [54]. In addition, osteopontin is also present in the phagolysosome of gut macrophages, and this was proposed to originate internally by transport of synthetic transport vesicles to the lysosome or externally by phagocytosis [54]. Thus, such dynamic traffic of osteopontin between cytoplasmic vesicles and external space might be quite similar to that observed in the several proteins localized in endometrial epithelium in the role of blastocyst implantation. Heparin-binding epidermal growth factor-like growth factor (HB-EGF) is expressed in a cyclic-dependent manner by luminal epithelial cells of the human endometrium, in response to both estrogen and progesterone [55]. HB-EGF is expressed at increasing amounts in the secretory-phase endometrium and is considered to be important for blastocyst implantation. Although the transmembrane form of HB-EGF may play an important role in cell adhesion and cell migration [56], maternal HB-EGF possibly binds to the blastocyst through a juxtacrine mechanism involving the EGF receptor, ErbB4. Thus, HB-EGF may not be involved in direct binding between blastocyst and endometrial cells.

### 3.6. Blastocyst’s EV Uptake in EECs after Interaction of Blastocyst Fibronectin with two Peptidases, EVs Derived from EECs and ES cells, and Blastocyst Attachment

In the human endometrium, a drastic change in exocytosis occurs; its peak coincides with the blastocyst arrival, whereas endocytosis ceases with implantation [35]. EVs are released into the uterine lumen to interact with blastocysts, possibly via their uptake. Thus, understanding EVs’ internalization and subsequent transfer of their cargo is essential for human blastocyst implantation.

It is speculated that EVs undergo fusion with MVBs in recipient cells to release their cargo. Recently, Joshi et al. [57] supported this speculation using green fluorescent protein and electron microscopy, revealing the release of EVs’ cargo in endosomes/lysosomes. They showed that internalized EVs fuse the membrane close to MVBs, composed of late endosomes and lysosomes in an acidification-dependent manner, resulting in cargo exposure to the cytosol. Localization of EVs in endosomes and lysosomes confirmed their uptake via endocytosis, but direct fusion of EVs with the plasma membrane was also noted to be possible [57]. Thus, EVs can enter cells via endocytosis and/or fusion.

Exosomes secreted by cancer cells carry a variety of molecules; therefore, they are used as delivery systems for tumor malignancy. An acidic microenvironment is a key factor in increasing exosome release and entry into melanoma cells [58]. It has also been argued that viruses may have adopted existing EV-mediated communication pathways for their infection [59].

The main characteristic of EVs is the enclosure and transfer of molecules with a lipid bilayer [60]. We have proposed that Aps, such as APN and APA, are not only present in plasma membranes, but also in lysosomes in EECs [1]. In addition, it was reported that, while the subset of the endosomal/lysosomal proteins contain cell-surface peptidases, such as DPPIV/CD26 and APN/CD13, exosomes do not contain any lysosomal APs [28,61]. Therefore, it can be speculated that the cargo in which both APs are present comprises lysosomes and/or MVs, rather than exosomes. After the binding of fibronectin in blastocysts to both DPPIV/CD26 and APN/CD13 in MVs/lysosomes, the MV/lysosome cargo attaches to the blastocyst, and then the blastocysts adhere to the EECs (Figure 2). The MV/lysosome cargo is attached to the pinopodes on the surface epithelium. Given that pinopodes are known to be quite stable due to poor apocrine secretion, pinopodes might be appropriate locations for blastocyst attachment.

EVs derived from EECs and treated with both estradiol and progesterone were taken up by the first-trimester human trophoblast cell line HTR8/SV neo, and subsequently induced a rapid increase in the adhesive capacity of HTR8/SV neo [40]. This may imply that EVs derived from the endometrium may potentially affect the cell adhesion capacity of blastocysts, thereby contributing to the interaction of blastocysts and EECs for successful implantation. Moreover, EVs derived from endometrial cells have recently been suggested to act on trophectoderm cells to promote implantation [62].

It was also shown that EVs derived from the ES cells of preimplantation embryos play active roles in implantation [9]. Given that the preparation of EVs derived from ES cells was devoid of exosomes, but mainly included MVs, the size of the EVs ranged from approximately 350 to 800 nm in diameter. When E3.5 blastocysts were isolated from the uteri of mice postcoital day 3.5, followed by culturing with ES cell MVs, they showed enhanced migration, significantly forming outgrowths. On the other hand, MVs secreted by HTR8/SV neo trophoblasts (a first-trimester human trophoblast cell line) failed to activate migration [9]. MVs derived from ES cells were shown to transfer their cargo to the trophectoderm layer, leading to MV-mediated intercellular communication between ES cells and trophoblasts, which positively affected the ability of trophoblasts to undergo implantation [9]. This was further supported by the fact that injecting E3.5 blastocysts with MVs isolated from ES cells resulted in a significant enhancement of blastocyst implantation by placing the blastocysts into the uteri of mice [9].

Both fibronectin and laminin were reported to be involved in the attachment and outgrowth of blastocysts in vitro; attachment and outgrowth of mouse blastocysts on tissue culture plates was significantly increased when the culture plates were individually precoated with fibronectin and laminin [63]. Given that fibronectin [6] and laminin [64] are present in human embryos during implantation, it is likely that they contribute to blastocyst attachment and outgrowth via MVs derived from ES cells, as proposed by Desrochers et al. [9].

Implantation is initiated within the microenvironment of uterine fluid, which contains a rich array of nutrients, proteins, lipids, and other molecules, arising from the endometrium stimulated by selective signal transduction from the blood side and, in the conception cycle, secretions of MVs from the blastocyst. In humans, initial attachment of the blastocyst occurs to the glycocalyx, as mentioned above [7]. EV bodies released from the endometrium (and possibly the embryo) are present in uterine fluid. It is, thus, reasonable that EVs provide an alternative means of exchange between blastocyst and endometrial epithelial cells for implantation. Therefore, we proposed that lysosomes are some of these EVs.

## 4. Conclusions

Implantation of the human blastocyst is a biological paradox that cannot be easily explained [65]. It is unclear how two epithelial cells (the trophectoderm cells of blastocysts and EECs) can make contact through their apical membranes [66]. This review provides a holistic overview of novel findings about the interaction of the blastocyst with two peptidases (DPPIV/CD26 and APN/CD13) in MVs/lysosomes derived from EECs, wherein the MV/lysosome cargo attached to the blastocyst undergoes back-fusion at the MVBs, possibly on the surface of pinopode cells, for implantation (Figure 2) [57]. Furthermore, the review discussed the possibility of blastocysts being involved in implantation via MVs secreted from ES cells. Thus, the microenvironment within the uterine cavity is critical during the final preimplantation stages of blastocyst development and for the successful establishment of pregnancy. Basic and clinical elucidation of the steroidal regulation and the function of APs in EVs/lysosomes for blastocyst implantation is necessary in the near future.

## Figures and Tables

**Figure 1 ijms-22-13479-f001:**
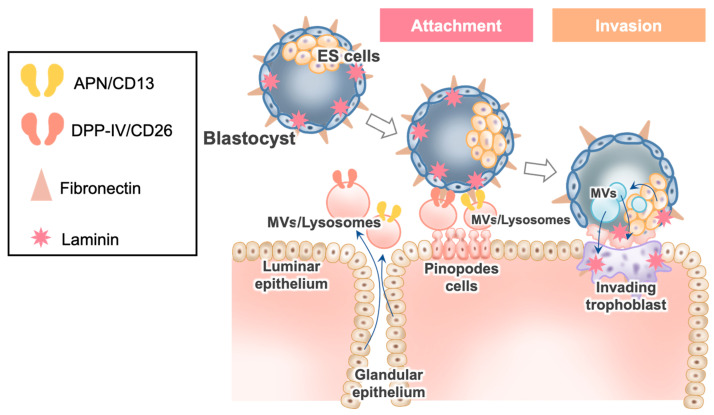
Hypothetical scheme for the possible involvement of APN/CD13 and DPPIV/CD26 expressed in MVs/lysosomes in blastocyte adhesion to EECs.

**Figure 2 ijms-22-13479-f002:**
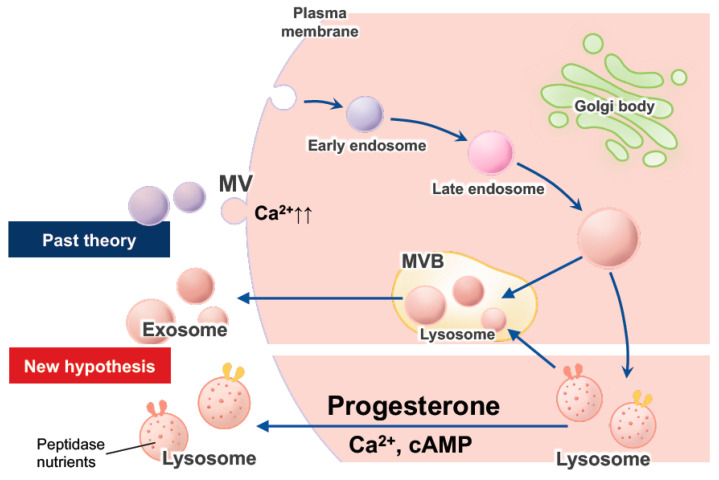
Schematic representation of the dynamics of the endolysosomal system and exocytosis of lysosomes.

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
