# Peer review of "Recent Insights into Human Endometrial Peptidases in Blastocyst Implantation via Shedding of Microvesicles"

_ijms, 2021, doi:10.3390/ijms222413479_

Round 1
Reviewer 1 Report
In the manuscript of Yoshihara et al the role of the plasma membrane and lysosome membrane dipeptidyl peptidase IV (DPPIV) and aminopeptidase N (APN) from the endometrial epithelial cells (EECs) in the interaction with the blastocyst during embryo implantation is analyzed, emphasizing the participation of extracellular vesicles (EVs) secreted by the endometrial glandular epithelium. They propose the interesting hypothesis that lysosomes are some of these EVs.
This review analyzes an interesting topic. Actualized information is integrated and the manuscript is well written. However, some aspects must be taken into consideration:
1.- The images of Figures 1 and 2 and changed. That of figure 2 corresponds with the legend of figure 1 and vice versa. It is important to situate adequately the references to figure 1 in the text when figure 1 is indicated in line 54 after the following asseveration “Increasing evidence has also shown that EVs of maternal and/or embryonal origin participate in the blastocyst–endometrial interactions that are critical to implantation (Figure 1)” appear that the image used in this figure is correct and probably the origin of this confusion.
2.- It is recommended that in the introduction, it must be established that the hypothesis that lysosomes are secreted as EVs will be proposed and analyzed. Lines 77 to 79 are confused considering EVs and lysosomes as homologs (EVs/lysosomes).
3.- Figure 1 and reference 6 indicate that EVs containing DPP-IV or APN are derived from the glandular epithelium and it is important to discriminate the role of glandular epithelium from the role of the endometrial epithelium in the text.
4.- Please analyze if there is an alternative mechanism of blastocyst attachment independent of fibronectin and/or independent of the mediation by EVs.
5.- The weakness of the hypothesis postulated in this manuscript must be analyzed.
Minor:
Please define the following abbreviations APs and LVs (in figure 2 legend).
Line 150; “After ovulation, intracellular Ca2+ and cyclic monophosphate (cAMP) levels are elevated (in which cells or tissue?).
Line 188, “apocrine excretion of lysosomes”, Excretion or secretion?
Sometimes the abbreviation EVs is used but others EV is also used, please unify.
Please reconsider this subtitle “3.6. Blastocyst uptake in EECs after interaction of blastocyst fibronectin with two peptidases in EVs” is not clear
Please change “HTR8” (lines 326 y 327) to “HTR8/SV neo” as this name is used in line 338.
Author Response
In the manuscript of Yoshihara et al the role of the plasma membrane and lysosome membrane dipeptidyl peptidase IV (DPPIV) and aminopeptidase N (APN) from the endometrial epithelial cells (EECs) in the interaction with the blastocyst during embryo implantation is analyzed, emphasizing the participation of extracellular vesicles (EVs) secreted by the endometrial glandular epithelium. They propose the interesting hypothesis that lysosomes are some of these EVs. This review analyzes an interesting topic. Actualized information is integrated and the manuscript is well written. However, some aspects must be taken into consideration.
- The images of Figures 1 and 2 and changed. That of figure 2 corresponds with the legend of figure 1 and vice versa. It is important to situate adequately the references to figure 1 in the text when figure 1 is indicated in line 54 after the following asseveration “Increasing evidence has also shown that EVs of maternal and/or embryonal origin participate in the blastocyst–endometrial interactions that are critical to implantation (Figure 1)” appear that the image used in this figure is correct and probably the origin of this confusion.
Answer:
Thank you for your correction. We amend the legend to Figures and appropriately changed the site of Figure 1.
- It is recommended that in the introduction, it must be established that the hypothesis that lysosomes are secreted as EVs will be proposed and analyzed. Lines 77 to 79 are confused considering EVs and lysosomes as homologs (EVs/lysosomes).
Answer:
Thank you for your kind instruction. We added the description in Introduction.
- Figure 1 and reference 6 indicate that EVs containing DPP-IV or APN are derived from the glandular epithelium and it is important to discriminate the role of glandular epithelium from the role of the endometrial epithelium in the text.
Answer:
Thank you for your comment. According to your instruction, we changed endometrial epithelium to glandular epithelium in the text.
- Please analyze if there is an alternative mechanism of blastocyst attachment independent of fibronectin and/or independent of the mediation by EVs.
Answer:
Thank you for your suggestion. We analyzed some alternative mechanisms of blastocyst attachment independent of fibronectin. We added the corresponding sentences in the last section of 3.5. Regarding the mechanism of blastocyst attachment independent of the mediation by EVs, we also made some comments on current concept of uterine microenvironment at which implantation is initiated in the last section of 3.6.
- The weakness of the hypothesis postulated in this manuscript must be analyzed.
Answer:
Thank you for your comment. As you pointed out, we just discussed the hypothesis, and it must be necessary to confirm these mechanisms with experiments in near future. We added this description in conclusion and believed that basic and clinical elucidation of the steroidal regulation and the function of APs in EVs/lysosomes for blastocyst implantation is necessary in near future.
Minor:
- Please define the following abbreviations APs and LVs (in figure 2 legend).
Answer:
Thank you for your comment. We amended them appropriately.
- Line 150; “After ovulation, intracellular Ca2+ and cyclic monophosphate (cAMP) levels are elevated (in which cells or tissue?).
Answer:
Thank you for your comment. We added “in EECs and ESCs” in the text.
- Line 188, “apocrine excretion of lysosomes”, Excretion or secretion?
Answer:
Thank you for your comment. We changed it to secretion.
- Sometimes the abbreviation EVs is used but others EV is also used, please unify.
Answer:
Thank you for your comment. We changed them appropriately.
- Please reconsider this subtitle “3.6. Blastocyst uptake in EECs after interaction of blastocyst fibronectin with two peptidases in EVs” is not clear
Answer:
Thank you for your comment. We changed it to “Blastocyst’s EVs uptake in EECs after interaction of blastocyst fibronectin with two peptidases”.
- Please change “HTR8” (lines 326 y 327) to “HTR8/SV neo” as this name is used in line 338.
Answer:
Thank you for your comment. We changed them appropriately.
Reviewer 2 Report
This is a very comprehensive review on the literature pertaining to the importance of aminopeptidases in human embryo implantation. The authors extensively review the mechanisms by which aminopeptidases could be externalized by endometrial cells, and the potential for binding by embryo-associated extracellular matrix. Overall it is well organized, informative, and well written.
Author Response
Reviewer 2
This is a very comprehensive review on the literature pertaining to the importance of aminopeptidases in human embryo implantation. The authors extensively review the mechanisms by which aminopeptidases could be externalized by endometrial cells, and the potential for binding by embryo-associated extracellular matrix. Overall it is well organized, informative, and well written.
Answer:
Thank you for your comment. We would like to confirm these proposals with basic and clinical researches in the near future.
Reviewer 3 Report
The manuscript by Yoshihara et al. is on a very interesting topic, with a focus on human endometrial peptidases in the context of embryo implantation. Overall, the Review is well written and clearly structured. However, large parts of the manuscript are not directly related to the topic contained in the title and the abstract (e.g., decidualization). Is there any data describing changes in peptidase expression in decidualized cells? For sure, there are more peptidases and peptidase inhibitors described to play a role during the preimplantation phase in humans and other mammals. The EVs and MVs could be integrated in the title. EVs have been described in various mammals to play a role in conceptus implantation.
Lines 39-40: I understand that the authors do not want to describe the histology of the endometrium in detail. But this way it is a bit oversimplified.
Paragraph 3.2: Where is the relation to APs?
Paragraph 3.3: I cannot find the description how FN is interacting with EVs’ molecules.
Heading 3.6 sounds a bit weird, I guess the authors wanted to say Blastocyst’ EVs uptake in EECs…?
Line 325: EECs?
Author Response
Reviewer 3
The manuscript by Yoshihara et al. is on a very interesting topic, with a focus on human endometrial peptidases in the context of embryo implantation. Overall, the Review is well written and clearly structured. However, large parts of the manuscript are not directly related to the topic contained in the title and the abstract (e.g., decidualization). Is there any data describing changes in peptidase expression in decidualized cells? For sure, there are more peptidases and peptidase inhibitors described to play a role during the preimplantation phase in humans and other mammals. The EVs and MVs could be integrated in the title. EVs have been described in various mammals to play a role in conceptus implantation.
Answer:
Thank you for your comments. We changed the title to add the concept of this review according to your kind instruction. We also added a description in text for readers to recognize our hypothesis as one of the proposed mechanisms of implantation.
- Lines 39-40: I understand that the authors do not want to describe the histology of the endometrium in detail. But this way it is a bit oversimplified.
Answer:
Thank you for your comment. We would like to discuss the detailed role of the endometrium, but we do not have enough data to support our hypothesis in each phase. We hope your understanding of the limitations of our proposals.
- Paragraph 3.2: Where is the relation to APs?
Answer:
In Paragraph 3.2 the exocytosis of lysosome in which APs are included, mainly due to progesterone-inducing change in membrane permeability was described.
- Paragraph 3.3: I cannot find the description how FN is interacting with EVs’ molecules.
Answer:
Would you please kindly understand it by overviewing sentences in 3.4 and 3.5.
- Heading 3.6 sounds a bit weird, I guess the authors wanted to say Blastocyst’ EVs uptake in EECs…?
Answer:
Thank you for your comment. We amended the heading.
Line 325: EECs?
Answer:
We appreciate your pointing out the mistake. We amended the word to EECs.
Round 2
Reviewer 1 Report
The manuscript has been improved and it is acceptable for publication.
Only two minor observations must be considered:
Line 322: please change “laminar epithelial cells” by “luminal epithelial cells”
Line 330-331. Reconsider the title of section 3.6, because an uptake of EV´s derived from the blastocyst by EECs is not described, an alternative subtitle could be: “EVs derived from EECs and ES cells and blastocyst attachment”.
Author Response
The manuscript has been improved and it is acceptable for publication.
Only two minor observations must be considered:
Answer:
Thank you for your reviewing. We really appreciate your helps.
- Line 322: please change “laminar epithelial cells” by “luminal epithelial cells”
Answer:
Thank you for your correction. We changed the text appropriately.
- Line 330-331. Reconsider the title of section 3.6, because an uptake of EV´s derived from the blastocyst by EECs is not described, an alternative subtitle could be: “EVs derived from EECs and ES cells and blastocyst attachment”.
Answer:
Thank you for your comment. We changed the subtitle to “EVs derived from EECs and ES cells and blastocyst attachment”.
Reviewer 3 Report
The authors addressed or clarified most of my comments and concerns.
Author Response
The authors addressed or clarified most of my comments and concerns.
Answer:
Thank you for your reviewing. We really appreciate your helps.